# Tree-Structured Reinforcement Learning for Sequential Object Localization

**Zequn Jie[1], Xiaodan Liang[2], Jiashi Feng[1], Xiaojie Jin[1], Wen Feng Lu[1], Shuicheng Yan[1]**
[1] National University of Singapore, Singapore
[2] Carnegie Mellon University, USA

## Abstract

Existing object proposal algorithms usually search for possible object regions over multiple locations and scales *separately*, which ignore the interdependency among different objects and deviate from the human perception procedure. To incorporate global interdependency between objects into object localization, we propose an effective Tree-structured Reinforcement Learning (Tree-RL) approach to sequentially search for objects by fully exploiting both the current observation and historical search paths. The Tree-RL approach learns multiple searching policies through maximizing the long-term reward that reflects localization accuracies over all the objects. Starting with taking the entire image as a proposal, the Tree-RL approach allows the agent to sequentially discover multiple objects via a tree-structured traversing scheme. Allowing multiple near-optimal policies, Tree-RL offers more diversity in search paths and is able to find multiple objects with a single feed-forward pass. Therefore, Tree-RL can better cover different objects with various scales which is quite appealing in the context of object proposal. Experiments on PASCAL VOC 2007 and 2012 validate the effectiveness of the Tree-RL, which can achieve comparable recalls with current object proposal algorithms via much fewer candidate windows.

## 1   Introduction

Modern state-of-the-art object detection systems [1, 2] usually adopt a two-step pipeline: extract a set of class-independent object proposals at first and then classify these object proposals with a pre-trained classifier. Existing object proposal algorithms usually search for possible object regions over dense locations and scales separately [3, 4, 5]. However, the critical correlation cues among different proposals (*e.g.*, relative spatial layouts or semantic correlations) are often ignored. This in fact deviates from the human perception process — as claimed in [6], humans do not search for objects within each local image patch separately, but start with perceiving the whole scene and successively explore a small number of regions of interest via sequential attention patterns. Inspired by this observation, extracting one object proposal should incorporate the global dependencies of proposals by considering the cues from the previous predicted proposals and future possible proposals jointly.

In this paper, in order to fully exploit global interdependency among objects, we propose a novel Tree-structured Reinforcement Learning (Tree-RL) approach that learns to localize multiple objects sequentially based on both the current observation and historical search paths. Starting from the entire image, the Tree-RL approach sequentially acts on the current search window either to refine the object location prediction or discover new objects by following a learned policy. In particular, the localization agent is trained by deep RL to learn the policy that maximizes a long-term reward for localizing all the objects, providing better global reasoning. For better training the agent, we

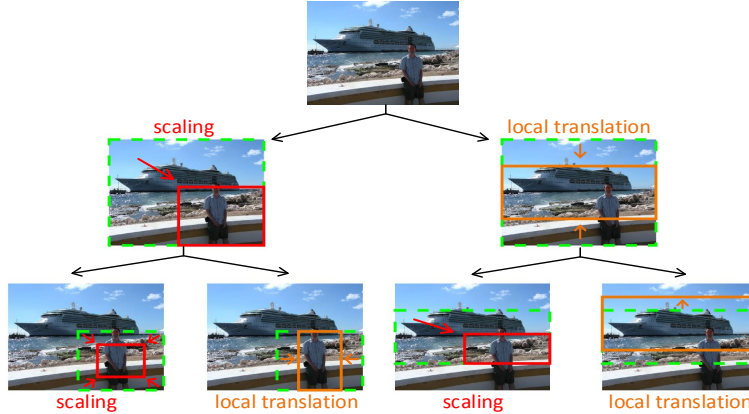

Figure 1: Illustration of Tree-RL. Starting from the whole image, the agent recursively selects the best actions from both action groups to obtain two next windows for each window. Red and orange solid windows are obtained by taking scaling and local translation actions, respectively. For each state, green dashed windows are the initial windows before taking actions, which are result windows from the last level.

propose a novel reward stimulation that well balances the exploration of uncovered new objects and refinement of the current one for quantifying the localization accuracy improvements.

The Tree-RL adopts a tree-structured search scheme that enables the agent to more accurately find objects with large variation in scales. The tree search scheme consists of two branches of pre-defined actions for each state, one for locally translating the current window and the other one for scaling the window to a smaller one. Starting from the whole image, the agent recursively selects the best action from each of the two branches according to the current observation (see Fig. 1). The proposed tree search scheme enables the agent to learn multiple near-optimal policies in searching multiple objects. By providing a set of diverse near-optimal policies, Tree-RL can better cover objects in a wide range of scales and locations.

Extensive experiments on PASCAL VOC 2007 and 2012 [7] demonstrate that the proposed model can achieve a similar recall rate as the state-of-the-art object proposal algorithm RPN [5] yet using a significantly smaller number of candidate windows. Moreover, the proposed approach also provides more accurate localizations than RPN. Combined with the Fast R-CNN detector [2], the proposed approach also achieves higher detection mAP than RPN.

## 2 Related Work

Our work is related to the works which utilize different object localization strategies instead of sliding window search in object detection. Existing works trying to reduce the number of windows to be evaluated in the post-classification can be roughly categorized into two types, *i.e.*, object proposal algorithms and active object search with visual attention.

Early object proposal algorithms typically rely on low-level image cues, *e.g.*, edge, gradient and saliency [3, 4, 8]. For example, Selective Search [9] hierarchically merges the most similar segments to form proposals based on several low-level cues including color and texture; Edge Boxes [4] scores a set of densely distributed windows based on edge strengths fully inside the window and outputs the high scored ones as proposals. Recently, RPN [5] utilizes a Fully Convolutional Network (FCN) [10] to densely generate the proposals in each local patch based on several pre-defined "anchors" in the patch, and achieves state-of-the-art performance in object recall rate. Nevertheless, object proposal algorithms assume that the proposals are independent and usually perform window-based classification on a set of reduced windows individually, which may still be wasteful for images containing only a few objects.

Another type of works attempts [11, 12, 13, 14] to reduce the number of windows with an active object detection strategy. Lampert *et al.* [15] proposed a branch-and-bound approach to find the highest scored windows while only evaluating a few locations. Alexe *et al.* [11] proposed a context driven active object searching method, which involves a nearest-neighbor search over all the training

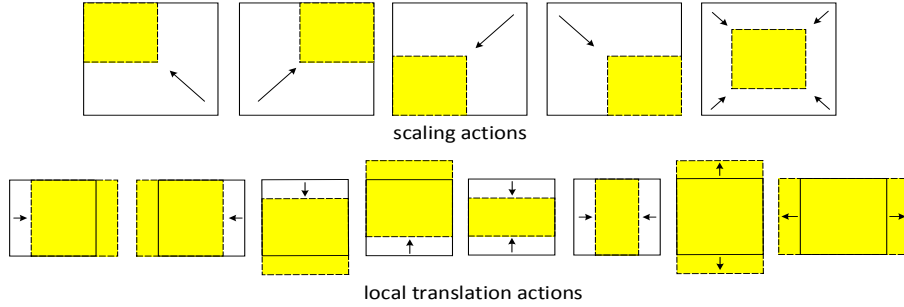

Figure 2: Illustration of the five scaling actions and eight local translation actions. Each yellow window with dashed lines represents the next window after taking the corresponding action.

images. Gonzeles-Garcia *et al.* [12] proposed an active search scheme to sequentially evaluate selective search object proposals based on spatial context information.

Visual attention models are also related to our work. These models are often leveraged to facilitate the decision by gathering information from previous steps in the sequential decision making vision tasks. Xu *et al.* [16] proposed an attention model embedded in recurrent neural networks (RNN) to generate captions for images by focusing on different regions in the sequential word prediction process. Minh *et al.* [17] and Ba *et al.* [18] also relied on RNN to gradually refine the focus regions to better recognize characters.

Perhaps [19] and [20] are the closest works to ours. [19] learned an optimal policy to localize a single object through deep Q-learning. To handle multiple objects cases, it runs the whole process starting from the whole image multiple times and uses an inhibition-of-return mechanism to manually mark the objects already found. [20] proposed a top-down search strategy to recursively divide a window into sub-windows. Then similar to RPN, all the visited windows serve as "anchors" to regress the locations of object bounding boxes. Compared to them, our model can localize multiple objects in a single run starting from the whole image. The agent learns to balance the exploration of uncovered new objects and the refinement of covered ones with deep Q-learning. Moreover, our top-down tree search does not produce "anchors" to regress the object locations, but provides multiple near-optimal search paths and thus requires less computation.

## 3 Tree-Structured Reinforcement Learning for Object Localization

### 3.1 Multi-Object Localization as a Markov Decision Process

The Tree-RL is based on a Markov decision process (MDP) which is well suitable for modeling the discrete time sequential decision making process. The localization agent sequentially transforms image windows within the whole image by performing one of pre-defined actions. The agent aims to maximize the total discounted reward which reflects the localization accuracy of all the objects during the whole running episode. The design of the reward function enables the agent to consider the trade-off between further refinement of the covered objects and searching for uncovered new objects. The actions, state and reward of our proposed MDP model are detailed as follows.

**Actions:** The available actions of the agent consist of two groups, one for scaling the current window to a sub-window, and the other one for translating the current window locally. Specifically, the scaling group contains five actions, each corresponding to a certain sub-window with the size 0.55 times as the current window (see Fig. 2). The local translation group is composed of eight actions, with each one changing the current window in one of the following ways: horizontal moving to left/right, vertical moving to up/down, becoming shorter/longer horizontally and becoming shorter/longer vertically, as shown in Fig. 2, which are similar to [19]. Each local translation action moves the window by 0.25 times of the current window size. The next state is then deterministically obtained after taking the last action. The scaling actions are designed to facilitate the search of objects in various scales, which cooperate well with the later discussed tree search scheme in localizing objects in a wide range of scales. The translation actions aim to perform successive changes of visual focus, playing an important role in both refining the current attended object and searching for uncovered new objects.

**States:** At each step, the state of MDP is the concatenation of three components: the feature vector of the current window, the feature vector of the whole image and the history of taken actions. The features of both the current window and the whole image are extracted using a VGG-16 [21] layer CNN model pre-trained on ImageNet. We use the feature vector of layer "fc6" in our problem. To accelerate the feature extraction, all the feature vectors are computed on top of pre-computed feature maps of the layer "conv5_3" after using ROI Pooling operation to obtain a fixed-length feature representation of the specific windows, which shares the spirit of Fast R-CNN. It is worth mentioning that the global feature here not only provides context cues to facilitate the refinement of the currently attended object, but also allows the agent to be aware of the existence of other uncovered new objects and thus make a trade-off between further refining the attended object and exploring the uncovered ones. The history of the taken actions is a binary vector that tells which actions have been taken in the past. Therefore, it implies the search paths that have already been gone through and the objects already attended by the agent. Each action is represented by a 13-d binary vector where all values are zeros except for the one corresponding to the taken action. 50 past actions are encoded in the state to save a full memory of the paths from the start.

**Rewards:** The reward function $r(s, a)$ reflects the localization accuracy improvements of all the objects by taking the action $a$ under the state $s$. We adopt the simple yet indicative localization quality measurement, Intersection-over-Union (IoU) between the current window and the ground-truth object bounding boxes. Given the current window $w$ and a ground-truth object bounding box $g$, IoU between $w$ and $g$ is defined as $\text{IoU}(w, g) \triangleq \text{area}(w \cap g)/\text{area}(w \cup g)$. Assuming that the agent moves from state $s$ to state $s'$ after taking the action $a$, each state $s$ has an associated window $w$, and there are $n$ ground-truth objects $g_1 \ldots g_n$, then the reward $r(s, a)$ is defined as follows:

$$r(s, a) = \max_{1 \leq i \leq n} \text{sign}(\text{IoU}(w', g_i) - \text{IoU}(w, g_i)). \tag{1}$$

This reward function returns $+1$ or $-1$. Basically, if any ground-truth object bounding box has a higher IoU with the next window than the current one, the reward of the action moving from the current window to the next one is $+1$, and $-1$ otherwise. Such binary rewards reflect more clearly which actions can drive the window towards the ground-truths and thus facilitate the agent's learning. This reward function encourages the agent to localize any objects freely, without any limitation or guidance on which object should be localized at that step. Such a free localization strategy is especially important in a multi-object localization system for covering multiple objects by running only a single episode starting from the whole image.

Another key reward stimulation $+5$ is given to those actions which cover any ground-truth objects with an IoU greater than 0.5 for the first time. For ease of explanation, we define $f_{i,t}$ as the hit flag of the ground-truth object $g_i$ at the $t^{th}$ step which indicates whether the maximal IoU between $g_i$ and all the previously attended windows $\{w^j\}_{j=1}^{t}$ is greater than 0.5, and assign $+1$ to $f_{i,t}$ if $\max_{1 \leq j \leq t} \text{IoU}(w_j, g_i)$ is greater than 0.5 and $-1$ otherwise. Then supposing the action $a$ is taken at the $t^{th}$ step under state $s$, the reward function integrating the first-time hit reward can be written as follows:

$$r(s, a) = \begin{cases} +5, & \text{if } \max_{1 \leq i \leq n} (f_{i,t+1} - f_{i,t}) > 0 \\ \max_{1 \leq i \leq n} \text{sign}(\text{IoU}(w', g_i) - \text{IoU}(w, g_i)), & \text{otherwise.} \end{cases} \tag{2}$$

The high reward given to the actions which hit the objects with an IoU $> 0.5$ for the first time avoids the agent being trapped in the endless refinement of a single object and promotes the search for uncovered new objects.

## 3.2 Tree-Structured Search

The Tree-RL relies on a tree structured search strategy to better handle objects in a wide range of scales. For each window, the actions with the highest predicted value in both the scaling action group and the local translation action group are selected respectively. The two best actions are both taken to obtain two next windows: one is a sub-window of the current one and the other is a nearby window to the current one after local translation. Such bifurcation is performed recursively by each window starting from the whole image in a top-down fashion, as illustrated in Fig. 3. With tree search, the agent is enforced to take both scaling action and local translation action simultaneously at

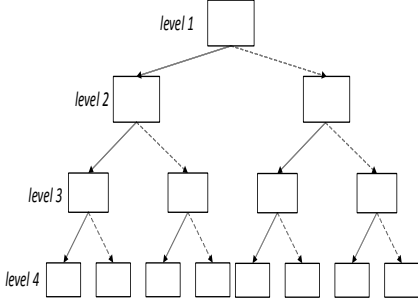

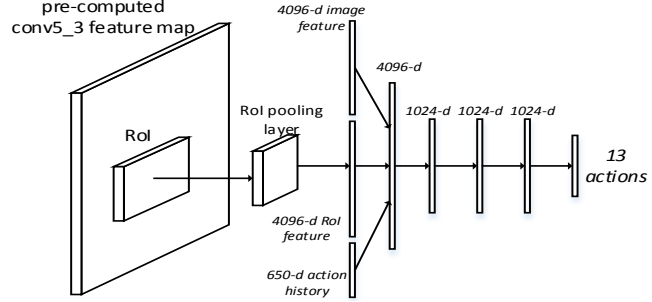

Figure 3: Illustration of the top-down tree search. Starting from the whole image, each window recursively takes the best actions from both action groups. Solid arrows and dashed arrows represent scaling actions and local translation actions, respectively.

Figure 4: Illustration of our Q-network. The regional feature is computed on top of the pre-computed "conv5_3" feature maps extracted by VGG-16 pre-trained model. It is concatenated with the whole image feature and the history of past actions to be fed into an MLP. The MLP predicts the estimated values of the 13 actions.

each state, and thus travels along multiple near-optimal search paths instead of a single optimal path. This is crucial for improving the localization accuracy for objects in different scales. Because only the scaling actions significantly change the scale of the attended window while the local translation actions almost keep the scale the same as the previous one. However there is no guarantee that the scaling actions are often taken as the agent may tend to go for large objects which are easier to be covered with an IoU larger than 0.5, compared to scaling the window to find small objects.

## 3.3  Deep Q-learning

The optimal policy of maximizing the sum of the discounted rewards of running an episode starting from the whole image is learned with reinforcement learning. However, due to the high-dimensional continuous image input data and the model-free environment, we resort to the Q-learning algorithm combined with the function approximator technique to learn the optimal value for each state-action pair which generalizes well to unseen inputs. Specifically, we use the deep Q-network proposed by [22, 23] to estimate the value for each state-action pair using a deep neural network. The detailed architecture of our Q-network is illustrated in Fig. 4. Please note that similar to [23], we also use the pre-trained CNN as the regional feature extractor instead of training the whole hierarchy of CNN, considering the good generalization of the CNN trained on ImageNet [24].

During training, the agent runs sequential episodes which are paths from the root of the tree to its leafs. More specifically, starting from the whole image, the agent takes one action from the whole action set at each step to obtain the next state. The agent's behavior during training is $\epsilon$-greedy. Specifically, the agent selects a random action from the whole action set with probability $\epsilon$, and selects a random action from the two best actions in the two action groups (*i.e.* scaling group and local translation group) with probability $1 - \epsilon$, which differs from the usual exploitation behavior that the single best action with the highest estimated value is taken. Such exploitation is more consistent with the proposed tree search scheme that requires the agent to take the best actions from both action groups. We also incorporate a replay memory following [23] to store the experiences of the past episodes, which allows one transition to be used in multiple model updates and breaks the short-time strong correlations between training samples. Each time Q-learning update is applied, a mini batch randomly sampled from the replay memory is used as the training samples. The update for the network weights at the $i^{th}$ iteration $\theta_i$ given transition samples $(s, a, r, s')$ is as follows:

$$\theta_{i+1} = \theta_i + \alpha(r + \gamma \max_{a'} Q(s', a'; \theta_i) - Q(s, a; \theta_i)) \nabla_{\theta_i} Q(s, a; \theta_i), \tag{3}$$

where $a'$ represents the actions that can be taken at state $s'$, $\alpha$ is the learning rate and $\gamma$ is the discount factor.

## 3.4  Implementation Details

We train a deep Q-network on VOC 2007+2012 trainval set [7] for 25 epochs. The total number of training images is around 16,000. Each epoch is ended after performing an episode in each training

Table 1: Recall rates (in %) of single optimal search path RL with different numbers of search steps and under different IoU thresholds on VOC 07 testing set. We only report 50 steps instead of 63 steps as the maximal number of steps is 50.

| # steps | large/small | IoU=0.5 | IoU=0.6 | IoU=0.7 |
|---------|-------------|---------|---------|---------|
| 31 | large | 62.2 | 53.1 | 40.2 |
| 31 | small | 18.9 | 15.6 | 11.2 |
| 31 | all | 53.8 | 45.8 | 34.5 |
| 50 | large | 62.3 | 53.2 | 40.4 |
| 50 | small | 19.0 | 15.8 | 11.3 |
| 50 | all | 53.9 | 45.9 | 34.8 |

Table 2: Recall rates (in %) of Tree-RL with different numbers of search steps and under different IoU thresholds on VOC 07 testing set. 31 and 63 steps are obtained by setting the number of levels in Tree-RL to 5 and 6, respectively.

| # steps | large/small | IoU=0.5 | IoU=0.6 | IoU=0.7 |
|---------|-------------|---------|---------|---------|
| 31 | large | 78.9 | 69.8 | 53.3 |
| 31 | small | 23.2 | 12.5 | 4.5 |
| 31 | all | 68.1 | 58.7 | 43.8 |
| 63 | large | 83.3 | 76.3 | 61.9 |
| 63 | small | 39.5 | 28.9 | 15.1 |
| 63 | all | 74.8 | 67.0 | 52.8 |

image. During $\epsilon$-greedy training, $\epsilon$ is annealed linearly from 1 to 0.1 over the first 10 epochs. Then $\epsilon$ is fixed to 0.1 in the last 15 epochs. The discount factor $\gamma$ is set to 0.9. We run each episode with maximal 50 steps during training. During testing, using the tree search, one can set the number of levels of the search tree to obtain the desired number of proposals. The replay memory size is set to 800,000, which contains about 1 epoch of transitions. The mini batch size in training is set to 64. The implementations are based on the publicly available Torch7 [25] platform on a single NVIDIA GeForce Titan X GPU with 12GB memory.

## 4 Experimental Results

We conduct comprehensive experiments on PASCAL VOC 2007 and 2012 testing sets of detection benchmarks to evaluate the proposed method. The recall rate comparisons are conducted on VOC 2007 testing set because VOC 2012 does not release the ground-truth annotations publicly and can only return a detection mAP (mean average precision) of the whole VOC 2012 testing set from the online evaluation server.

**Tree-RL vs Single Optimal Search Path RL:** We first compare the performance in recall rate between the proposed Tree-RL and a single optimal search path RL on PASCAL VOC 2007 testing set. For the single optimal search path RL, it only selects the best action with the highest estimated value by the deep Q-network to obtain one next window during testing, instead of taking two best actions from the two action groups. As for the exploitation in the $\epsilon$-greedy behavior during training, the agent in the single optimal path RL always takes the action with the highest estimated value in the whole action set with probability $1 - \epsilon$. Apart from the different search strategy in testing and exploitation behavior during training, all the actions, state and reward settings are the same as Tree-RL. Please note that for Tree-RL, we rank the proposals in the order of the tree depth levels. For example, when setting the number of levels to 5, we have 1+2+4+8+16=31 proposals. The recall rates of the single optimal search path RL and Tree-RL are shown in Table 1 and Table 2, respectively. It is found that the single optimal search path RL achieves an acceptable recall with a small number of search steps. This verifies the effectiveness of the proposed MDP model (including reward, state and actions setting) in discovering multiple objects. It does not rely on running multiple episodes starting from the whole image like [19] to find multiple objects. It is also observed that Tree-RL outperforms the single optimal search path RL in almost all the evaluation scenarios, especially for large objects[1]. The only case where Tree-RL is worse than the single optimal search path RL is the recall of small objects within 31 steps at IoU threshold 0.6 and 0.7. This may be because the agent performs a breadth-first-search from the whole image, and successively narrows down to a small region. Therefore, the search tree is still too shallow (*i.e.* 5 levels) to accurately cover all the small objects using 31 windows. Moreover, we also find that recalls of the single optimal search path RL become stable with a few steps and hardly increase with the increasing of steps. In contrast, the recalls of Tree-RL keep increasing as the levels of the search tree increase. Thanks to the multiple diverse near-optimal search paths, a better coverage of the whole image in both locations and scales is achieved by Tree-RL.

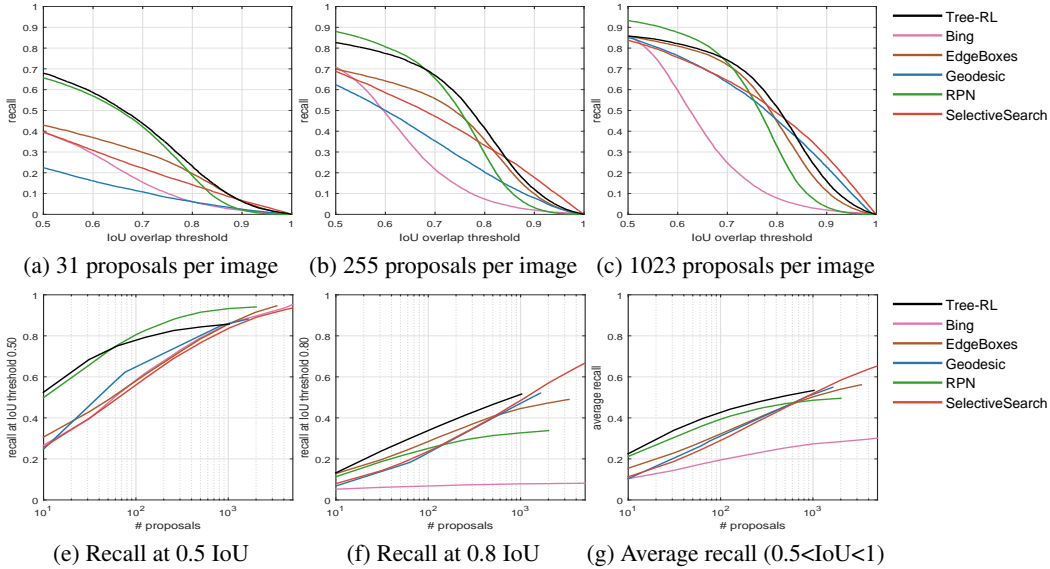

(a) 31 proposals per image　(b) 255 proposals per image　(c) 1023 proposals per image

(e) Recall at 0.5 IoU　(f) Recall at 0.8 IoU　(g) Average recall (0.5<IoU<1)

Figure 5: Recall comparisons between Tree-RL and other state-of-the-art methods on PASCAL VOC 2007 testing set.

**Recall Comparison to Other Object Proposal Algorithms:** We then compare the recall rates of the proposed Tree-RL and the following object proposal algorithms: BING [3], Edge Boxes [4], Geodesic Object Proposal [26], Selective Search [9] and Region Proposal Network (RPN) [5] (VGG-16 network trained on VOC 07+12 trainval) on VOC 2007 testing set. All the proposals of other methods are provided by [27]. Fig. 5 (a)-(c) show the recall when varying the IoU threshold within the range [0.5,1] for different numbers of proposals. We set the number of levels in Tree-RL to 5, 8 and 10 respectively to obtain the desired numbers of proposals. Fig. 5 (e)-(g) demonstrate the recall when changing the number of proposals for different IoU thresholds. It can be seen that Tree-RL outperforms other methods including RPN significantly with a small number of proposals (*e.g.* 31). When increasing the number of proposals, the advantage of Tree-RL over other methods becomes smaller, especially at a low IoU threshold (*e.g.* 0.5). For high IoU thresholds (*e.g.* 0.8), Tree-RL stills performs the best among all the methods. Tree-RL also behaves well on the average recall between IoU 0.5 to 1 which is shown to correlate extremely well with detector performance [27].

**Detection mAP Comparison to Faster R-CNN:** We conduct experiments to evaluate the effects on object detection of the proposals generated by the proposed Tree-RL. The two baseline methods are RPN (VGG-16) + Fast R-CNN (ResNet-101) and Faster R-CNN (ResNet-101). The former one trains a Fast R-CNN detector (ResNet-101 network) on the proposals generated by a VGG-16 based RPN to make fair comparisons with the proposed Tree-RL which is also based on VGG-16 network. The latter one, *i.e.* Faster-RCNN (ResNet-101), is a state-of-the-art detection framework integrating both proposal generation and object detector in an end-to-end trainable system which is based on ResNet-101 network. Our method, Tree-RL (VGG-16) + Fast R-CNN (ResNet-101) trains a Fast R-CNN detector (ResNet-101 network) on the proposals generated by the VGG-16 based Tree-RL. All the Fast R-CNN detectors are fine-tuned from the publicly released ResNet-101 model pre-trained on ImageNet. The final average pooling layer and the 1000-d fc layer of ResNet-101 are replaced by a new fc layer directly connecting the last convolution layer to the output (classification and bounding box regression) during fine-tuning. For Faster-RCNN (ResNet-101), we directly use the reported results in [28]. For the other two methods, we train and test the Fast R-CNN using the top 255 proposals. Table 3 and Table 4 show the average precision of 20 categories and mAP on PASCAL VOC 2007 and 2012 testing set, respectively. It can be seen that the proposed Tree-RL combined with Fast R-CNN outperforms two baselines, especially the recent reported Faster R-CNN (ResNet-101) on the detection mAP. Considering the fact that the proposed Tree-RL relies on only VGG-16 network which is much shallower than ResNet-101 utilized by Faster R-CNN in proposal generation, the proposed Tree-RL is able to generate high-quality object proposals which are effective when used in object detection.

Table 3: Detection results comparison on PASCAL VOC 2007 testing set.

| method | aero | bike | bird | boat | bottle | bus | car | cat | chair | cow | table | dog | horse | mbike | person | plant | sheep | sofa | train | tv | mAP |
|---|---|---|---|---|---|---|---|---|---|---|---|---|---|---|---|---|---|---|---|---|---|
| RPN (VGG-16)+ Fast R-CNN (ResNet-101) | 77.7 | 82.7 | 77.4 | 68.5 | 54.7 | 85.5 | 80.0 | 87.6 | 60.7 | 83.2 | 71.8 | 84.8 | 85.1 | 75.6 | 76.9 | 52.0 | 76.8 | 79.1 | 81.1 | 73.9 | 75.8 |
| Faster R-CNN (ResNet-101) [28] | 79.8 | 80.7 | 76.2 | 68.3 | 55.9 | 85.1 | 85.3 | 89.8 | 56.7 | 87.8 | 69.4 | 88.3 | 88.9 | 80.9 | 78.4 | 41.7 | 78.6 | 79.8 | 85.3 | 72.0 | 76.4 |
| Tree-RL (VGG-16)+ Fast R-CNN (ResNet-101) | 78.2 | 82.4 | 78.0 | 69.3 | 55.4 | 86.0 | 79.3 | 88.4 | 60.8 | 85.3 | 74.0 | 85.7 | 86.3 | 78.2 | 77.2 | 51.4 | 76.4 | 80.5 | 82.2 | 74.5 | 76.6 |

Table 4: Detection results comparison on PASCAL VOC 2012 testing set.

| method | aero | bike | bird | boat | bottle | bus | car | cat | chair | cow | table | dog | horse | mbike | person | plant | sheep | sofa | train | tv | mAP |
|---|---|---|---|---|---|---|---|---|---|---|---|---|---|---|---|---|---|---|---|---|---|
| RPN (VGG-16)+ Fast R-CNN (ResNet-101) | 86.9 | 83.3 | 75.6 | 55.4 | 50.8 | 79.2 | 76.9 | 92.8 | 48.8 | 79.0 | 57.2 | 90.2 | 85.4 | 82.1 | 79.4 | 46.0 | 77.0 | 66.4 | 83.3 | 66.0 | 73.1 |
| Faster R-CNN (ResNet-101) [28] | 86.5 | 81.6 | 77.2 | 58.0 | 51.0 | 78.6 | 76.6 | 93.2 | 48.6 | 80.4 | 59.0 | 92.1 | 85.3 | 84.8 | 80.7 | 48.1 | 77.3 | 66.5 | 84.7 | 65.6 | 73.8 |
| Tree-RL (VGG-16)+ Fast R-CNN (ResNet-101) | 85.9 | 79.3 | 77.1 | 62.1 | 53.4 | 77.8 | 77.4 | 90.1 | 52.3 | 79.2 | 56.2 | 88.9 | 84.5 | 80.8 | 81.1 | 51.7 | 77.3 | 66.9 | 82.6 | 68.5 | 73.7 |

**Visualizations:** We show the visualization examples of the proposals generated by Tree-RL in Fig. 6. As can be seen, within only 15 proposals (the sum of level 1 to level 4), Tree-RL is able to localize the majority of objects with large or middle sizes. This validates the effectiveness of Tree-RL again in its ability to find multiple objects with a small number of windows.

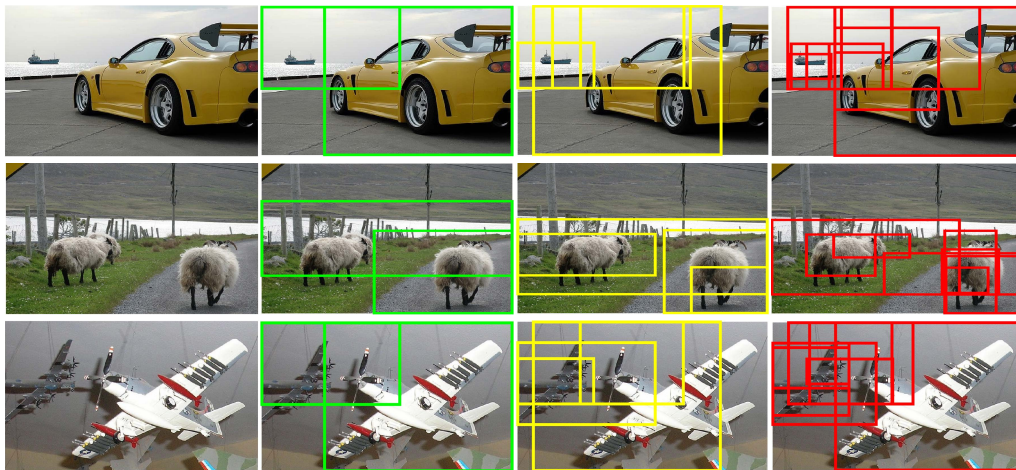

Figure 6: Examples of the proposals generated by Tree-RL. We only show the proposals of level 2 to level 4. Green, yellow and red windows are generated by the 2nd, 3rd and 4th level respectively. The 1st level is the whole image.

## 5  Conclusions

In this paper, we proposed a novel Tree-structured Reinforcement Learning (Tree-RL) approach to sequentially search for objects with the consideration of global interdependency between objects. It follows a top-down tree search scheme to allow the agent to travel along multiple near-optimal paths to discovery multiple objects. The experiments on PASCAL VOC 2007 and 2012 validate the effectiveness of the proposed Tree-RL. Briefly, Tree-RL is able to achieve a comparable recall to RPN with fewer proposals and has higher localization accuracy. Combined with Fast R-CNN detector, Tree-RL achieves comparable detection mAP to the state-of-the-art detection system Faster R-CNN (ResNet-101).

## Acknowledgment

The work of Jiashi Feng was partially supported by National University of Singapore startup grant R-263-000-C08-133 and Ministry of Education of Singapore AcRF Tier One grant R-263-000-C21-112.

## Footnotes

[1]Throughout the paper, large objects are defined as those containing more than 2,000 pixels. The rest are small objects.

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
