[Reviews · NeurIPS 2016]

Reviewer 1

Summary

This paper proposes a method for generating object proposals by using a sequential search strategy that is trained with reinforcement learning (Q-learning specifically). The paper has two major contributions: one, at test time, instead of taking just one action at each step, it takes two actions (one best action each from two classes of actions) and bifurcates. This allows the model to actually predict multiple objects and deal with the associated multimodality better. Two, the reward signal is carefully designed and while not imposing any order, provides rewards both when the localization of an object improves and when the model latches on to a new object. The recall achieved by the method seems impressive, and when plugged into the fast r-cnn improves over RPN by about 2 points.

Qualitative Assessment

I liked the ideas present in the paper. This is the first sequential search strategy I have seen that tries to output all objects in an image, and does not impose any arbitrary and hard-to-justify order on the boxes during training. I have a minor clarification, and then some ways of strengthening the experimental section, which to me would be the difference between a poster and an oral. A point of clarification: It seems that the ranking of the proposals output is simply the depth of the tree at which they are discovered. Why is this a good ranking? Is there anything in the training that encourages the model to discover objects quickly? I can think of some intuitive explanation but it would be good for the authors to provide an explanation in the paper. Proposals are a fast moving subfield of computer vision and this paper would do well to add a few more experiments: 1) Comparison to more proposal methods. In particular, I would want comparisons to MCG[1] and DeepMask[2], both of which are significantly better than, eg. edge boxes. MCG proposals are also available online 2) Comparison on COCO. COCO captures a much larger range of scale variations especially when it comes to small objects and methods like RPN typically do not fare very well. I would like to see how this fares. In addition, if the authors envisage this as a proposal method, then it should generalize to object categories that the network has not seen during training. It would be good to see if this is indeed true and if so, to what extent; COCO has a few categories that are not present either in PASCAL or Imagenet, so I would be curious how this method fares when trained on PASCAL and tested on COCO It would also be nice to see what the detection AP looks like for higher overlap thresholds: this method seems to improve recall for higher IU so it might provide larger gains under more stringent evaluation metrics.

Confidence in this Review

3-Expert (read the paper in detail, know the area, quite certain of my opinion)


Reviewer 2

Summary

This paper proposes a novel tree based reinforcement learning approach to sequentially propose the object bounding boxes in a deep Q learning network framework. Empirically, this leads to competitive region proposal results (recall @ nboxes, recall @ iou) and state of the art object detection results (mAP) on PASCAL07 test set.

Qualitative Assessment

Interesting paper applying reinforcement learning to object proposals. One thing I noticed from the recall plots (Figure 5 e and f) is that the authors didn't evaluate the performance beyond 1500 proposals. So it's hard to compare apples-to-apples with other baseline proposal methods how the recall compares beyond that 1500 number. For example, other baselines such as selective search, edge boxes, bing show recall close to 1.0 at around 4000 boxes with iou threshold of 0.5. However, the proposed method seems to saturate quite early on (upon eye-balling to extrapolat on my own) and seems unlikely to have high recall beyond the 1500 number authors evaluated. Regardless of the results, it'd make the paper scientifically more solid if the authors report the results beyond 1500 boxes.

Confidence in this Review

2-Confident (read it all; understood it all reasonably well)


Reviewer 3

Summary

The authors propose a reinforcement learning approach for speeding up the search object scales and locations. When proposing new objects, it relies on a tree-structured search scheme that scales (down) or translates the search window in each step. In this it is somewhat motivated by visual attention models. The approach overall speeds up and improves the proposal generation and leads to state-of-the-art results at reduced computational costs when implemented together with standard object detectors.

Qualitative Assessment

The paper presents a small interesting idea that is well presented and analyzed. I like the idea of following a formal representation of a visual attention model in the proposal generation, rather than a brute force dense CNN-based classification. I also consider the deep Q learning optimization approach to be interesting. Overall, this study may be of interest to NIPS attendees.

Confidence in this Review

1-Less confident (might not have understood significant parts)


Reviewer 4

Summary

The paper presents an object proposal algorithms which exploits reinforcement learning modeled as a MDP and a novel tree-structured search scheme that allows discovery of multiple objects in a single pass based on two seemingly well defined actions; translation and scale search. Combined with detection procedures, the proposed algorithm not only presents the improvement for the object proposal task but also shows effectiveness in the object detection task.

Qualitative Assessment

The basic idea of using a tree-structured search scheme with multiple policies to allow the discovery of multiple objects in a single pass is clever, straightforward and make sense. The approach is technically sound and novel, although the basic MDP model (e.g., 13 actions) seem to be derived from a previous work and be extended. The experiments are well set up proving the efficacy of the proposed method, however, I would like to see the detection case results of either only VGG or Resnet is used in a baseline and the proposed algorithm. I also have an impression that authors state that the detection is better than state-of-the-arts because of their object proposal, however, the mAP reported in the paper (~%76) is somewhat lower than the currently existing best detection performances reported in the literature. Overall, the paper is very well written and may have an impact.

Confidence in this Review

2-Confident (read it all; understood it all reasonably well)


Reviewer 5

Summary

In this paper, to capture the interdependency among different objects, a tree structured reinforcement learning (Tree-RL) approach to sequentially search for object candidates is proposed. An entire image is a starting point and the agent sequentially discovers multiple objects via a tree-structured traversing scheme. The agent takes scaling action and local translation action simultaneously at each tree node, and travels along multiple near-optimal search paths learned by the deep Q learning framework. Experimental sections are designed to show three aspects: comparison between tree search and single path search, comparison of their algorithm to several object proposal algorithms and comparison of object detection frameworks with and without their object proposals.

Qualitative Assessment

- As a preprocessor, algorithms for object proposals are required to be time-efficient. All of algorithms [3,4,25,10,5] shown in Figure 5 reported their run-time. In this paper, there is no time measurement or complexity analysis while authors insist that their model is computationally efficient. Authors should compare other algorithms in terms of time-efficiency. - Authors might include more qualitative results in Figure 6. All of 3 examples contain 2 main objects. It would benefit if qualitative comparisons of their algorithm to other object proposals are added with cases that only one main object or more than 2 objects appear in images. - The motivations/justifications of their particular tree structures seem less clear. 1) Why translation and scaling actions are chosen for the left and right child respectively? Why not other variations are considered? 3) Could some randomisation in the node split benefit in terms of generalisation or other strategies? - The technical contribution/novelty of the paper is marginal compared to [18]. Except the section 3.2 (Tree-structured Search), section 3 including the definition of actions/states/rewards and optimization method (Deep Q learning) is similar to [18,22]. Another difference to [18] is, as authors pointed out, [18] runs the whole process multiple times to detect objects while this paper runs the whole process once sequentially; This comparison can be made clearer in experiments in terms of time-efficiency and more qualitative/quantitative results.

Confidence in this Review

2-Confident (read it all; understood it all reasonably well)


Reviewer 6

Summary

This paper describes a Q-learning combined with deep learning approach to object detection and localization. In particular this paper introduces a tree-based reinforcement learning that sequentially allows to determine which region of the image is the best candidate to represent an object bounding box. The verification is done on VOC2007/2012 and the results are competitive with others state of art detectors.

Qualitative Assessment

The paper presents an interesting improvement to the usual RL. Several points could be clarified however: - the desired number of proposal is a parameter. While it was experimentally determined from testing it would be interesting to formulate the problem of finding of proposals in a more formal manner - is there any changes between agent tree-RL while searching for different objects?

Confidence in this Review

2-Confident (read it all; understood it all reasonably well)